# BALLAST: Bayesian Active Learning with Look-ahead Amendment for Sea-drifter Trajectories

**Rui-Yang Zhang**
Lancaster University
r.zhang26@lancaster.ac.uk

**Henry B. Moss**
Lancaster University
henry.moss@lancaster.ac.uk

**Lachlan C. Astfalck**
The University of Western Australia
lachlan.astfalck@uwa.edu.au

**Edward J. Cripps**
The University of Western Australia
edward.cripps@uwa.edu.au

**David S. Leslie**
Lancaster University
d.leslie@lancaster.ac.uk

## Abstract

We introduce a formal experimental design methodology for guiding the placement of drifters to infer ocean currents. The majority of drifter placement campaigns either follow standard 'space-filling' designs or relatively ad-hoc expert opinions, which could be significantly improved on by appealing to statistical experimental design. Drifter observations follow a Lagrangian structure as drifters are advected through the Eulerian vector field. It is, therefore, important to consider the likely future trajectories of placed drifters to compute the utility of candidate measurement locations. We present BALLAST: Bayesian Active Learning with Look-ahead Amendment for Sea-drifter Trajectories, a novel light-weight, nonmyopic policy amendment for drifter trajectory data that can be appended to any myopic policy. Our numerical studies suggest benefits in incorporating BALLAST into the sequential placement strategies of drifters.

## 1  Introduction

Understanding and predicting ocean currents are of vital importance to mapping the flow of heat, nutrients, pollutants and sediments in the ocean [3, 9]. Ocean currents are inferred from a plurality of measurement devices, such as fixed-location buoys, satellites and free-floating buoys, known as drifters [11]. Free-floating drifters are being increasingly used due to their ability to sample both spatial and temporal flow properties and remain relatively affordable as compared to other measurement devices [15].

Standard experiment designs consider measurement devices that operate at a single location at a single time. Drifters are different: advected by the background vector field, drifters proceed to move following the Lagrangian flow of the ocean currents and collect data at multiple locations at multiple times (see the orange observation trajectories and the extended red observations in Figure 1). Drifter measurements thus require nontrivial policies for experiment designs and sequential experiment designs (SED) where observations are made incrementally — allowing efficient dynamic experimentation by adapting to the results of previous measurements. However, the majority of existing drifter placement campaigns either follow standard 'space-filling' designs [19] or relatively

Workshop on Bayesian Decision-making and Uncertainty, 38th Conference on Neural Information Processing Systems (NeurIPS 2024).

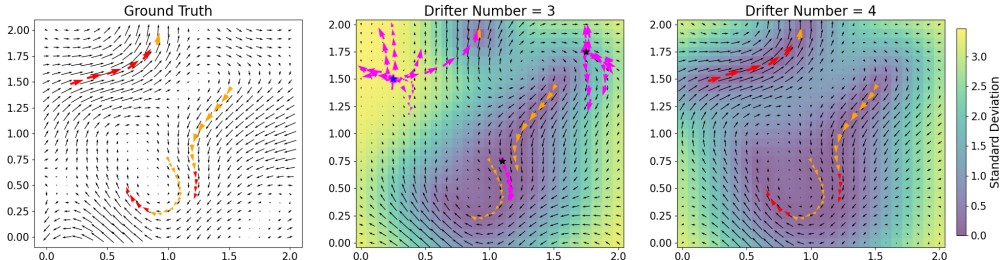

Figure 1: An example run with Max Var BALLASTs method with 10 sampled look-aheads. (Left) The ground truth field. (Mid) The estimated field using three placed drifters with the predictive standard deviation as the heat map. Trajectories of the three currently placed drifters are in orange. The three stars indicate three candidate locations for the next placement, with purple arrows indicating the 10 look-ahead trajectories sampled from our current model of the underlying flow. The blue star is the selected location for the next placement. (Right) The estimated field using four placed drifters with new observations in red.

ad-hoc expert opinions [21]. To the best of our knowledge, methodology for a formal statistical design of drifter allocation strategies does not exist.

Standard (S)ED policies only consider the observation made immediately at the selected observation point, making them *myopic* [4, 16] and are likely insufficient to guide the temporally extended Lagrangian observations associated with drifter placement designs. Policies that take into account potential future observations, often called *nonmyopic* policies, exist in the literature [8, 7, 23], but they focus on solving or approximating the Bellman equation considering the next several observations actively selected by the decision-maker (see [5] for a discussion). A practical sequential placement strategy for drifters should incorporate potential future trajectories while remaining computationally light-weight.

In this work, we introduce an experimental design strategy for drifters — Bayesian Active Learning with Look-ahead Amendment for Sea-drifter Trajectories (BALLAST) — that offers a *light-weight*, *nonmyopic* policy amendment by leveraging a vector-output Gaussian process model and the drifters' Lagrangian trajectory structure. We make three primary contributions: (i) we introduce sequential experimental design concepts to the problem of drifter placement, (ii) we propose a novel light-weight, nonmyopic SED policy amendment that can be easily appended to most existing myopic SED policies like maximum variance and maximum integrated variance [6], and (iii) we present the results of numerical simulations that showcase improved performance of our proposed policy over common alternatives.

## 2   Background

**Gaussian Processes**   A vector-output Gaussian process model is used here to model the 2-dimensional, stationary vector field that represents the ocean currents over a region. A vector-output Gaussian process (GP) $\{\boldsymbol{y}(x)\}_{x \in \mathcal{X}}$ with input space $\mathcal{X}$, output space $\mathbb{R}^D$ with $D > 1$, $\mathbb{R}^D$-valued mean function $\mu$ and $\mathbb{R}^{D \times D}$-valued covariance function $k$ is a continuous stochastic process fully denoted as $\boldsymbol{y}(\cdot) \sim \mathcal{GP}(\mu(\cdot), k(\cdot, \cdot))$ such that every finite collection of random variables in $\{\boldsymbol{y}(x)\}_{x \in \mathcal{X}}$ has a multivariate normal distribution [22, 1]. Normally, the mean function $\mu$ is set to be a constant-zero function, which will be the case here too. Details about regressions and predictions with a vector-output GP can be found in Appendix A.

**Experimental Design**   Experimental design considers the problem of choosing the optimal measurement $x^*$ given some existing data $\mathcal{D}$ and a model $g$ according to some utility function $U$ that calculates the expected quality of each possible measurement $x$ within the measurement set $R$ under the belief of our latest model $g$. Mathematically, this is given by $x^* = \text{argmax}_{x \in R} \mathbb{E}_g[U(x, g, \mathcal{D})]$. When we wish to make measurements *sequentially*, or adaptively, we will find an optimal measurement $x^*_{n+1}$ for each time $n$ based on existing data $\mathcal{D}_n$ and latest model $g_n$ according to $x^*_{n+1} = \text{argmax}_{x \in R} \mathbb{E}_{g_n}[U(x, g_n, \mathcal{D}_n)]$.

In this work, we will use a Gaussian process as our model due to its expressibility, conjugacy with data updates and inherent uncertainty quantification. The choice of utility functions is central to the effectiveness of an (S)ED. Two common choices of utility functions are the variance of our latest model $g_n$ at the candidate decision $x$ (yielding the *maximum variance policy*), and the total variance of the posterior distribution of our model after being updated by the candidate decision $x$ and its outcome $y$ (yielding the *maximum integrated variance policy*). See [6, 18] for comprehensive introductions to these designs.

## 3 Sequential Drifter Placement and BALLAST

The problem of sequential drifter placement can be formulated as follows. Consider a fixed region $R \subset \mathbb{R}^2$ of the ocean that we wish to understand its currents. Let $\boldsymbol{F}$ denote the true stationary vector field of the region $R$. We will place drifters sequentially into the region and use the observations to infer the true field $\boldsymbol{F}$, and the goal is to maximally reduce our uncertainty about $\boldsymbol{F}$ using a minimum number of drifters.

Assuming $n$ drifters are currently deployed, we have existing data $\mathcal{D}_n$ and latest model $g_n$. With a utility function choice $U$, the myopic allocation policy is to only consider the initial placement point from $R$, selected by

$$x^* = \underset{x \in R}{\operatorname{argmax}} \, \mathbb{E}_{g_n}[U(x, g_n, \mathcal{D}_n)]. \tag{1}$$

Recall that drifters are free-floating and will keep collecting data at different locations once placed. A myopic policy does not take into account the future trajectory of a drifter placed at $x$ and could favour locations that are apparently more promising yet uninformative overall — say the placed drifter leaves $R$ immediately or is stuck in a convergent region and so is, in effect, stationary. With myopic policies, the Lagrangian structure of the drifter observations is completely ignored.

### 3.1 BALLAST

We propose the BALLAST method, a nonmyopic policy amendment with look-aheads of drifter trajectories that is computationally expedient. While still considering only the initial placement point $x \in R$, we construct the look-ahead of a drifter placed at $x$ via a projection of $T$ steps with step size $\delta$ using a vector field $\hat{\boldsymbol{F}}$, denoted by $P_\delta^T(x, \hat{\boldsymbol{F}}) \in R^{T+1}$. Given $n$ existing drifters with data $\mathcal{D}_n$ and model $g_n$, the **BALLAST** method determines the next placement location by

$$x^* = \underset{x \in R}{\operatorname{argmax}} \, \mathbb{E}_{g_n}\left[U(P_\delta^T(x, \hat{\boldsymbol{F}}), g_n, \mathcal{D}_n)\right]. \tag{2}$$

We will denote the conversion from Equation (1) to Equation (2) as the **BALLAST amendment** to the base myopic policy. Various design choices of this amendment are discussed below.

**Look-ahead Projection** The trajectory of a drifter at $x$ inside the vector field $\hat{\boldsymbol{F}}$ can be approximated using numerical ordinary differential equation solvers. The simplest choice is the Euler method, which iteratively updates each step by $x_{(m+1)\delta} = x_{m\delta} + \delta\hat{\boldsymbol{F}}(x_{m\delta})$ for $m \in \mathbb{Z}^{\geq 0}$ where $\hat{\boldsymbol{F}}(x)$ denotes the value of the field $\hat{\boldsymbol{F}}$ at $x$. Other solvers such as the implicit Euler method can also be used [17].

**Step Size and Number** Caution is required when setting the step size $\delta$ and the step number $T$ to balance the trade-off between practicality and accuracy. A tiny step size $\delta$ may induce unwanted computational costs, yet too large a step size will produce poor approximations of the drifter dynamics and yield inaccurate predictions. For step number $T$, a small value will make the policy more myopic, which we wish to avoid, yet a large value will incur additional computational costs. Furthermore, the look-ahead is based on an estimated vector field $\hat{\boldsymbol{F}}$, which could be erroneous, so too long of a look-ahead might be misleading.

**Estimated Vector Field** Choice of the estimated vector field reflects our latest understanding of currents in the region $R$. As alluded to previously, the quality of the estimated vector field strongly influences the projection quality. Here, we propose two ways to construct the estimated field which

induce two versions of BALLAST. The first way is to choose, naively, the predictive mean $\mu_{\mathcal{D}_n}$ of the latest GP model $g_n$, that leads to the policy

$$x^* = \underset{x \in R}{\arg\max} \, \mathbb{E}_{g_n} \left[ U(P_\delta^T(x, \mu_{\mathcal{D}_n}), g_n, \mathcal{D}_n) \right] \tag{3}$$

which we call **BALLASTn**. The second way is to draw $K$ independent sample fields $\{\hat{\boldsymbol{F}}_k\}_{k=1}^K$ from the predictive distribution of the latest model $g_n$ and estimate the utility using the sample averages. Denote this second method by **BALLASTs**, with $\hat{\boldsymbol{F}}_k \sim g_n$ for $k = 1, 2, \ldots, K$, we have

$$x^* = \underset{x \in R}{\arg\max} \, \frac{1}{K} \sum_{k=1}^K \mathbb{E}_{g_n} \left[ U(P_\delta^T(x, \hat{\boldsymbol{F}}_k), g_n, \mathcal{D}_n) \right]. \tag{4}$$

One should note that BALLASTs and BALLASTn are different. For BALLASTs, when we take the limit $k \to \infty$, we get

$$\frac{1}{K} \sum_{k=1}^K U(P_\delta^T(x, \hat{\boldsymbol{F}}_k), g_n, \mathcal{D}_n) \to \mathbb{E}_{\hat{\boldsymbol{F}} \sim g_n} \left[ U(P_\delta^T(x, \hat{\boldsymbol{F}}), g_n, \mathcal{D}_n) \right]$$

whereas for BALLASTn we have

$$U(P_\delta^T(x, \mathbb{E}_{\hat{\boldsymbol{F}} \sim g_n}[\hat{\boldsymbol{F}}]), g_n, \mathcal{D}_n)$$

instead. Since $U \circ P_\delta^T$ is not a linear function in the field choice, the two quantities are not identical, showing that BALLASTs and BALLASTn are indeed different.

**Utility Function**   It is obvious from the formulation that the BALLAST methods are compatible with any utility function $U$. With the look-ahead amendment provided by BALLAST methods, we have observed noticeable improvements from utility choices such as maximum variance and maximum integrated variance, as supported by the simulations in the following section.

## 4   Simulation Results

We generate a ground truth vector field $\boldsymbol{F}$ from a vector-valued GP parameterised by the Helmholtz kernel of [2] with prespecified hyperparameters. A total of 8 drifters are sequentially placed with $T \times \delta = 5 \times 0.03$ unit time apart, and the observations provided by the drifters follow the ground truth field $\boldsymbol{F}$ with small additive observation noise. When a drifter leaves the region $R$, we terminate its motion. Eight placement policies are considered: uniform sampling, Sobol sequence [10], maximum variance, maximum variance with BALLASTn, maximum variance with BALLASTs using 100 sampled look-aheads, maximum integrated variance, maximum integrated variance with BALLASTn, and maximum integrated variance with BALLASTs using 100 sampled look-aheads. As demonstrated in Figure 1, we update the model and compute the mean squared error (MSE) of the predictive mean with each placement. Further details of the experiment are found in Appendix B.

Figure 2 shows experimental results over 100 random starting seeds and indicates improvements, regardless of the base utility functions, of the BALLAST amendments. The improvement is more significant as more drifters are placed. Furthermore, a simple policy like maximum variance after BALLAST amendments displays comparable performance to a more complicated myopic policy, the maximum integrated variance.

We have not carried out a thorough computational cost comparison, but for indicative timings using a standard laptop, one full run of the simulation with the maximum variance policy takes 15 seconds, while that with the BALLASTn takes around 20 seconds and that with the BALLASTs using 100 sampled look-aheads under simple parallelization of the samples takes around 1 minute 15 seconds. For operational drifter placement, these computational costs are likely to be insignificant.

## 5   Conclusion

In this paper, we introduce a formal experimental design of the sequential placement of drifters for ocean currents inference. Our numerical results showed that SED-based policies noticeably outperformed common ad-hoc or space-filling drifter allocation strategies, which are further outperformed

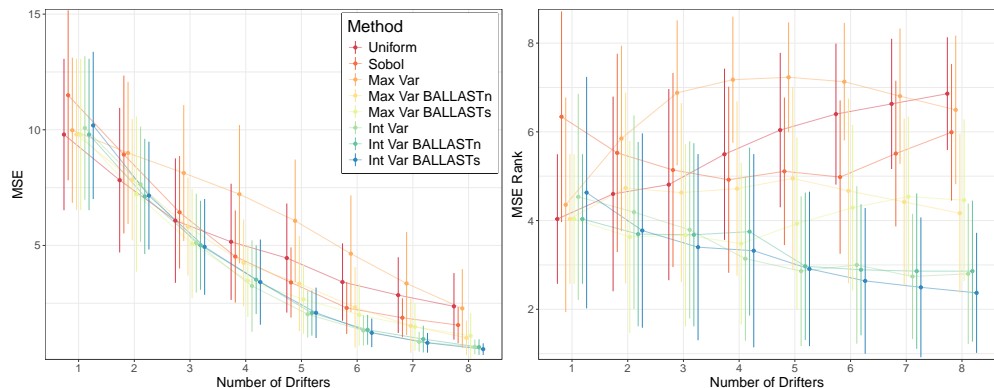

Figure 2: (Left) Mean MSE over the number of drifters placed. (Right) Mean rank (lower is better) of MSE over the number of drifters placed. One standard deviation radius error bounds are included.

by our proposed light-weight and nonmyopic BALLAST extensions. Potential further work includes considering time-varying ocean flows, for which more efficient sampling methods will be investigated [20], establishing theoretical guarantees for the non-myopic benefits of BALLAST, and incorporating BALLAST into a wider risk-averse [14] framework or incorporating information-theoretic framework for batch [12] or multi-fidelity designs [13].

## Acknowledgments and Disclosure of Funding

RZ thanks Ben Lowery for the help with accessing the compute cluster. RZ is supported by EPSRC funded STOR-i Center for Doctoral Training (grant no. EP/S022252/1). RZ, LA and EC are supported by the ARC ITRH for Transforming energy Infrastructure through Digital Engineering (TIDE), Grant No. IH200100009.

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

## A  Vector-Output Gaussian Process Details

For a $D$-vector-output GP model $\boldsymbol{y} \sim \mathcal{GP}(\mu, k_{\boldsymbol{\beta}})$ with kernel specified by hyperparameters $\boldsymbol{\beta}$, $N$ observations $\mathcal{D} = X \cup Y = \{x_i\}_{i=1}^N \cup \{\boldsymbol{y}_i\}_{i=1}^N$ for input $x_i$ and noisy output $\boldsymbol{y}_i = \boldsymbol{y}(x_i) + \boldsymbol{\varepsilon}, \boldsymbol{\varepsilon} \sim N_D(0, S)$ with $S = \sigma^2 I_D$, if we let $\boldsymbol{\theta} = (\boldsymbol{\beta}, \sigma)$ denote all the unknown parameters of $\boldsymbol{y}$, then the log-likelihood is given by

$$l(\boldsymbol{\theta}|\mathcal{D}) = -\frac{1}{2}\overline{\boldsymbol{y}}^T (K_{\boldsymbol{\beta}}(X, X) + \Sigma)^{-1}\overline{\boldsymbol{y}} - \frac{1}{2}\log|K_{\boldsymbol{\beta}}(X, X) + \Sigma| - \frac{ND}{2}\log 2\pi$$

where $\overline{y} \in \mathbb{R}^{ND}$ is the concatenation of observations $\{y_i\}_{i=1}^{N}$, $K_{\beta}(X, X) \in \mathbb{R}^{ND \times ND}$ with

$$K_{\beta}(X, X) = \begin{bmatrix} K_{1,1}(X,X) & K_{1,2}(X,X) & \dots & K_{1,D}(X,X) \\ K_{2,1}(X,X) & K_{2,2}(X,X) & \dots & K_{2,D}(X,X) \\ \vdots & \vdots & \dots & \vdots \\ K_{D,1}(X,X) & K_{D,2}(X,X) & \dots & K_{D,D}(X,X) \end{bmatrix}$$

and $K_{d,d'}(X, X) = \{k_{\beta}(x_i, x_j)\}_{i,j=1}^{N}$ for $d, d' = 1, 2, \dots, D$, and $\Sigma = S \otimes I_N$ with $\otimes$ denoting the Kronecker product.

Furthermore, the posterior predictive distribution $y(x^*)|\mathcal{D}$ of a test point $x^* \in \mathcal{X}$ after observing $\mathcal{D}$ is provided by

$$\boldsymbol{y}(x^*)|\boldsymbol{y}, x^*, X, \overline{\boldsymbol{y}}, \boldsymbol{\theta} \sim N(\mu_*(x^*), K_*(x^*, x^*))$$
$$\mu_*(x^*) = K_{\beta}(x^*, X)[K_{\beta}(X, X) + \Sigma]^{-1}\overline{\boldsymbol{y}}$$
$$K_*(x^*, x^*) = K_{\beta}(x^*, x^*) - K_{\beta}(x^*, X)[K_{\beta}(X, X) + \Sigma]^{-1}K_{\beta}(x^*, X)^T$$

where $K_{\beta}(x^*, X) \in \mathbb{R}^{D \times ND}$ is the concatenation of $D \times D$ matrix $K_{\beta}(x^*, x_j)$ where $j = 1, 2, \dots, N$.

## A.1 The Helmholtz kernel

The vector-output kernel of choice for this paper is the Helmholtz kernel of [2], which is a recently proposed model that can better capture the vector field structure using the Helmholtz decomposition. Denoting the 2D vector field by $\boldsymbol{F} = [u, v]^T$ with Helmholtz decomposition $\boldsymbol{F} = \text{grad}\boldsymbol{\Phi} + \text{rot}\boldsymbol{\Psi}$ for the potential function $\boldsymbol{\Phi}$ and the stream function $\boldsymbol{\Psi}$, the Helmholtz kernel imposes a GP prior to each of the two functions, i.e. $\boldsymbol{\Phi} \sim \mathcal{GP}(0, k_{\Phi})$, $\boldsymbol{\Psi} \sim \mathcal{GP}(0, k_{\Psi})$, which gives the overall kernel $\boldsymbol{F} \sim \mathcal{GP}(0, k_{\text{Helm}})$ with the kernel function

$$k_{\text{Helm}}(\boldsymbol{x}, \boldsymbol{x'}) = \begin{bmatrix} \partial^2_{x_1 x_1'} k_{\Phi}(\boldsymbol{x}, \boldsymbol{x'}) + \partial^2_{x_2 x_2'} k_{\Psi}(\boldsymbol{x}, \boldsymbol{x'}) & \partial^2_{x_1 x_2'} k_{\Phi}(\boldsymbol{x}, \boldsymbol{x'}) - \partial^2_{x_2 x_1'} k_{\Psi}(\boldsymbol{x}, \boldsymbol{x'}) \\ \partial^2_{x_2 x_1'} k_{\Phi}(\boldsymbol{x}, \boldsymbol{x'}) - \partial^2_{x_1 x_2'} k_{\Psi}(\boldsymbol{x}, \boldsymbol{x'}) & \partial^2_{x_2 x_2'} k_{\Phi}(\boldsymbol{x}, \boldsymbol{x'}) + \partial^2_{x_1 x_1'} k_{\Psi}(\boldsymbol{x}, \boldsymbol{x'}) \end{bmatrix}.$$

for $\boldsymbol{x}, \boldsymbol{x'} \in \mathbb{R}^2$ if we assume $\boldsymbol{\Phi}$ and $\boldsymbol{\Psi}$ are independent. The two kernels of $\boldsymbol{\Phi}, \boldsymbol{\Psi}$ are chosen to be squared exponential kernels with respective lengthscales $l_{\Phi}, l_{\Psi}$.

## B Additional Details and Plots of the Experiment

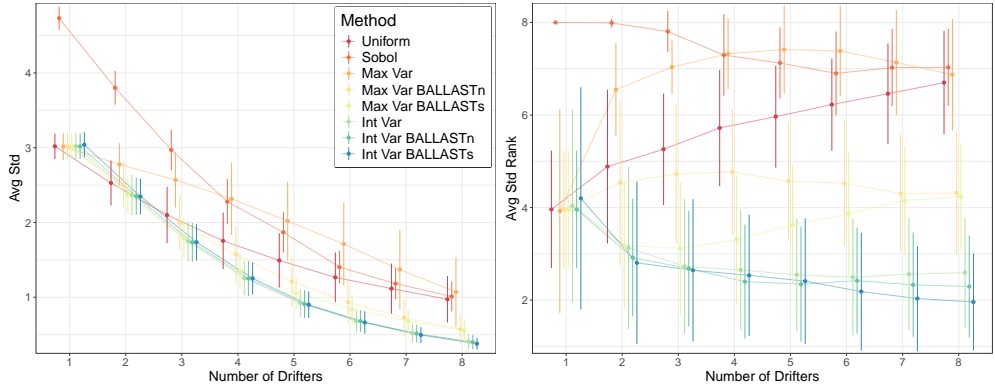

Figure 3: (Left) Mean average standard deviation over the number of drifters placed. (Right) Mean rank (lower is better) of average standard deviation over the number of drifters placed. One standard deviation radius error bounds are included.

The ground truth vector field $\boldsymbol{F}$ is generated using the Helmholtz kernel of [2], further explained in Appendix A.1, with known hyperparameters (lengthscales $l_{\Phi} = 0.7, l_{\Psi} = 0.5$ for $k_{\Phi}, k_{\Psi}$ respectively) over the region $R = [0, 2]^2$ and the square $25 \times 25$ grid $G$. A total of 8 drifters are sequentially

placed, $T\delta$ unit time apart, with $T = 5$ and $\delta = 0.03$, and the observations provided by the drifters follow the ground truth field $\boldsymbol{F}$ via an Euler discretization of step size $\delta = 0.03$ and additive centered Gaussian observation noise with variance $0.01$. The velocity of the drifter is taken to be the value of the vector field at the nearest block of the current position. When a drifter leaves the region $R$, we will terminate its motion. Eight placement policies are considered: uniform sampling, Sobol sequence, maximum variance, maximum variance with BALLASTs, maximum variance with BALLASTn using 100 samples, integrated variance, integrated variance with BALLASTs, and integrated variance with BALLASTs using 100 samples. The first drifters under all policies, except for the case of the Sobol sequence, are placed at a uniformly drawn location of $G$.

The model we use is the Helmholtz kernel with the true hyperparameters. With each placement, we update the model with all existing data and compute the mean squared error (MSE) of the predictive mean against the truth $\hat{\boldsymbol{F}}$ as well as the average standard deviation of the predictive distribution. A total of 100 iterations with different initial seeds are done for all eight policies, and the results are summarised in Tables 1 and 2.

Table 1: Mean MSE over 100 Runs for all eight policies across the number of drifters placed.

| | 1 | 2 | 3 | 4 | 5 | 6 | 7 | 8 |
|---|---|---|---|---|---|---|---|---|
| **Uniform** | 9.794 | 7.822 | 6.074 | 5.157 | 4.456 | 3.417 | 2.852 | 2.366 |
| **Sobol** | 11.491 | 8.930 | 6.433 | 4.521 | 3.399 | 2.298 | 1.872 | 1.556 |
| **Max Var** | 9.979 | 8.997 | 8.130 | 7.216 | 6.063 | 4.643 | 3.357 | 2.278 |
| **Max Var BALLASTn** | 9.794 | 7.840 | 5.810 | 4.244 | 3.351 | 2.325 | 1.524 | 1.006 |
| **Max Var BALLASTs** | 9.794 | 7.220 | 5.073 | 3.472 | 2.672 | 2.001 | 1.482 | 1.094 |
| **Int. Var** | 10.077 | 7.645 | 5.099 | **3.248** | **2.025** | 1.321 | 0.836 | 0.616 |
| **Int. Var BALLASTn** | 9.794 | **7.120** | 5.006 | 3.520 | 2.079 | 1.352 | 0.951 | 0.607 |
| **Int. Var BALLASTs** | 10.188 | 7.154 | **4.938** | 3.415 | 2.087 | **1.221** | **0.789** | **0.516** |

Table 2: Mean Average Standard Deviation over 100 Runs for all eight policies across the number of drifters placed.

| | 1 | 2 | 3 | 4 | 5 | 6 | 7 | 8 |
|---|---|---|---|---|---|---|---|---|
| **Uniform** | 3.019 | 2.528 | 2.100 | 1.754 | 1.492 | 1.266 | 1.116 | 0.974 |
| **Sobol** | 4.732 | 3.801 | 2.971 | 2.279 | 1.868 | 1.403 | 1.183 | 1.011 |
| **Max Var** | 3.016 | 2.777 | 2.570 | 2.317 | 2.020 | 1.713 | 1.370 | 1.070 |
| **Max Var BALLASTn** | 3.019 | 2.501 | 2.000 | 1.572 | 1.209 | 0.936 | 0.726 | 0.576 |
| **Max Var BALLASTs** | 3.019 | 2.373 | 1.785 | 1.351 | 1.058 | 0.839 | 0.680 | 0.551 |
| **Int. Var** | 3.022 | 2.367 | 1.747 | 1.256 | 0.921 | 0.681 | 0.519 | 0.408 |
| **Int. Var BALLASTn** | 3.019 | 2.351 | **1.735** | **1.250** | 0.911 | 0.681 | 0.515 | 0.400 |
| **Int. Var BALLASTs** | 3.040 | **2.348** | 1.737 | 1.253 | **0.900** | **0.663** | **0.496** | **0.379** |

We have also produced the mean average standard deviation summary and rank plots in Figure 3, similar to those in Figure 2 for MSE. We can notice very similar, positive results of the BALLAST methods.

