# OpenReview forum: "BALLAST: Bayesian Active Learning with Look-ahead Amendment for Sea-drifter Trajectories"
_NeurIPS.cc/2024/Workshop/BDU — NeurIPS BDU Workshop 2024 Poster_

### Official Review · Reviewer_oqc7 · 2024-09-14
**Review on "BALLAST: Bayesian Active Learning with Look-ahead Amendment for Sea-drifter Trajectories"**

**Rating:** 5
**Confidence:** 3

**Review:**

The paper presents a formal experimental design to optimize the placement of sea drifters used for measuring and inferring ocean currents. The main contribution of this paper is a lightweight Bayesian Active Learning-based drifter experimental design strategy, and the authors claim they are the first to give a formal statistical design of drifter allocation strategies. The numerical evaluation demonstrates the effectiveness of their methods.

Strength:
[1] The authors claim they are the first ones to give a formal statistical design of drifter allocation strategies.

[2] The results demonstrate the advantage of their methods over baselines in terms o MSE.

Weakness:

[1] The related works are lacking in the Introduction part. The authors should provide enough evidence to prove the exclusiveness of their work. For example, in the second paragraph of the Introduction, the authors mention the "space-filling" and relatively ad-hoc expert opinions, but they don't cite any existing references.

[2] Will the change of the first deployment of the drifter affect the final utility? The authors didn't mention how to optimally determine the first drifter placement and its effect on BALLAST.

[3] The experiments should be expanded to explore more factors (e.g., size of areas) on different methods.

---

### Official Review · Reviewer_HtqV · 2024-09-14
**The authors propose a new Bayesian active learning algorithm to determine the optimal placement of drifters for measuring ocean currents. The effectiveness of the proposed methodology is demonstrated in a simulated experiment.**

**Rating:** 5
**Confidence:** 3

**Review:**

## Strengths
- The use of Bayesian experimental design for the problem of drifter placement appears to be novel.
- The paper is well-written and easy to follow.

---
## Weaknesses
- A true non-myopic policy must consider the placement of future drifters and not just the trajectory followed by the next drifter. The authors address this difference in the following sentence in the introduction:
> A suitable policy for drifter placement should be non-myopic, in the sense that future observations from stochastic projections of drifter trajectories are considered, even if they do not go to the extent of solving the Bellman equation to consider future drifter placement.

The latter part of this sentence is unconvincing. Recent works (see, for e.g., [1]) have shown that fully non-myopic policies significantly outperform myopic ones, and the authors provide no discussion or reasoning behind their choice in this paper. I'm assuming it is for computational reasons, but this point has to be elucidated further.

- A comparison of BALLASTn and BALLASTs is missing in the text. What is the motivation behind introducing BALLASTs? The sample average is just an unbiased estimator of the predictive mean.
- I did not find that the claim
> Our numerical studies suggest substantial benefits in incorporating BALLAST into the sequential placement strategies of drifters.

in the abstract is sufficiently backed up by the simulation results. For example, in the plot on the left in Figure 2, using the maximum integrated variance utility function seems to yield more or less identical results regardless of whether or not BALLAST is used. The plot on the right only shows marginal improvement, and that too only for BALLASTs. Why would BALLASTs outperform BALLASTn? The authors do not provide adequate insights into the results of the experiments. In short, I feel that the numerical validation in the paper is insufficient to claim any substantial benefit for using BALLAST, and further simulations are necessary to make conclusions.

---
## Minor comments
- Figure 2 is difficult to read (too much going on). A table with the same information could help the reader compare methods more easily.
- The following sentence starting on line 26, which makes a claim about existing work, is lacking citations.
> However, the majority of existing drifter placement campaigns either follow standard ‘space-filling’ designs or relatively ad-hoc expert opinions.

---
## References
[1] Foster, A., Ivanova, D. R., Malik, I., and Rainforth, T. Deep adaptive design: Amortizing sequential Bayesian experimental design. In International Conference on Machine Learning, 2021.

---

### Decision · Program_Chairs · 2024-10-09

**Decision:**

Accept (Poster)

**Comment:**

Reviews for this paper are borderline, but their actual text is more positive than the scores. Reviewers praise the application, but complain about comprehensiveness of experiments. Given this is a workshop paper, I think these complains, while valid, are not sufficient to warrant rejection. I therefore opt to place this work on the "accept" side of the borderline paper threshold.